# Systematic proteome and proteostasis profiling in human Trisomy 21 fibroblast cells

Yansheng Liu [1], Christelle Borel[2], Li Li[3], Torsten Müller[1], Evan G. Williams[1], Pierre-Luc Germain[4], Marija Buljan[1], Tatjana Sajic[1], Paul J. Boersema[5], Wenguang Shao[1], Marco Faini[1], Giuseppe Testa[4,6], Andreas Beyer [3], Stylianos E. Antonarakis[2,7] & Ruedi Aebersold [1,8]

Down syndrome (DS) is mostly caused by a trisomy of the entire Chromosome 21 (Trisomy 21, T21). Here, we use SWATH mass spectrometry to quantify protein abundance and protein turnover in fibroblasts from a monozygotic twin pair discordant for T21, and to profile protein expression in 11 unrelated DS individuals and matched controls. The integration of the steady-state and turnover proteomic data indicates that protein-specific degradation of members of stoichiometric complexes is a major determinant of T21 gene dosage outcome, both within and between individuals. This effect is not apparent from genomic and transcriptomic data. The data also reveal that T21 results in extensive proteome remodeling, affecting proteins encoded by all chromosomes. Finally, we find broad, organelle-specific post-transcriptional effects such as significant downregulation of the mitochondrial proteome contributing to T21 hallmarks. Overall, we provide a valuable proteomic resource to understand the origin of DS phenotypic manifestations.

[1] Department of Biology, Institute of Molecular Systems Biology, ETH Zurich, 8093 Zurich, Switzerland. [2] Department of Genetic Medicine and Development, University of Geneva Medical School, and University Hospitals of Geneva, 1211 Geneva, Switzerland. [3] Cellular Networks and Systems Biology, University of Cologne, CECAD, University of Cologne, 50931 Cologne, Germany. [4] Department of Experimental Oncology, European Institute of Oncology, 20139 Milan, Italy. [5] Institute of Biochemistry, Department of Biology, ETH Zurich, 8093 Zurich, Switzerland. [6] Department of Oncology and Hemato-Oncology, University of Milan, 20122 Milan, Italy. [7] iGE3 Institute of Genetics and Genomics of Geneva, 1211 Geneva, Switzerland. [8] Faculty of Science, University of Zurich, 8057 Zurich, Switzerland. Christelle Borel and Li Li contributed equally to this work. Correspondence and requests for materials should be addressed to A.B. (email: andreas.beyer@uni-koeln.de) or to S.E.A. (email: Stylianos.Antonarakis@unige.ch) or to R.A. (email: aebersold@imsb.biol.ethz.ch)

Down syndrome (DS) is caused by the presence of a supernumerary chromosome 21 (Chr21). DS is among the most prevalent genetic disorders, occurring in ~1 in 750 newborns. The phenotypic manifestation of DS, which includes intellectual disability, congenital heart defects, and increased occurrence of Alzheimer's disease, is highly variable across individuals[1]. The molecular mechanisms underlying the phenotypic manifestations are still poorly understood. A simple working hypothesis suggests that human Chr21 genes display altered expression in individuals with T21 due to genome dosage imbalance. Those dosage-sensitive Chr21 genes showing increased expression by about 1.5-fold are expected to contribute to the DS phenotype, whereas dosage insensitive genes are not[1]. Recent transcript-centric studies were directed toward the identification of dosage-sensitive Chr21 genes[2–5]. Transcriptome profiles have been reported for various human tissues[4, 6] and mouse models trisomic for syntenic regions of Chr21[1,8].

Although the impact of T21 on the transcriptome has been relatively well studied, we lack understanding of how T21 affects the proteome. A review of recent work integrating genomic, transcriptomic, and proteomic data has revealed that the effect of copy number variations (CNVs) on protein levels is much more complex than previously anticipated[9]. For example, studies of cancer cells and yeast show that many CNVs resulting in messenger RNA (mRNA) changes only weakly influence corresponding protein levels[9–12]. Since DS is the prototype of a chromosome gain CNV, herein we study the proteomic effects of T21.

Skin fibroblasts have been used to study gene dosage at the transcript level for T21 since 1979[13], but to date no proteomic T21 data have been available for this cell type. We herein study the proteomes from skin fibroblasts from 11 genetically unrelated DS individuals and 11 controls. Additionally, we include fibroblasts from an extremely rare pair of monozygotic (MZ) twins discordant for T21[14]. This unique design of our study allows us to compare the proteomic impact of T21 in a genetically identical background to the impact observed in genetically distinct individuals.

The primary aim of our study is to analyze T21-associated proteome changes. However, the expected relatively small protein level fold changes caused by the 1.5-fold gene dosage increase

poses a technical challenge for current proteomics technologies. Indeed, previous protein measurements of DS cells, mainly facilitated by antibody-based methods, only identified marginal overexpression of a few proteins coded on Chr21[15–17]. To circumvent this limitation, we apply SWATH mass spectrometry (SWATH-MS), a recently developed, massively parallel protein targeting technique, which features high accuracy and reproducibility in protein quantification[18–20]. Additionally, we apply pulsed SILAC (pSILAC)[21–23] to quantify the turnover of corresponding proteins and to further reveal mechanisms of proteome adaptation. The integrative data reveal two principal responses to the perturbation caused by T21: the first is protein-specific degradation, maintaining stoichiometric stability in heteromeric protein complexes both within and across individuals, and the second is organelle-specific proteostasis.

## Results

**A high-quality quantitative T21 proteome data set.** To accurately quantify the proteomic changes in DS, we used SWATH-MS[18] to analyze the proteome of fetal skin primary fibroblasts derived from 11 DS and 11 unrelated, sex and age-matched normal control (N) individuals[4] (hereafter, "unrelated samples", Fig. 1a, b; Supplementary Data 1), each with two experimental replicates. Additionally, we analyzed a pair of MZ twins discordant for T21 (namely T2N and T1DS) for which transcriptomic data was previously reported[14]. For each twin two independent cell cultures, three experimental replicates were analyzed, resulting in six proteomic profiles per individual (Supplementary Fig. 1).

The resulting SWATH-MS data were processed with Open-SWATH[24] and further aligned with TRIC[25] by targeted signal extraction, using mass spectrometric query parameters for 10,000 human proteins as prior information[26]. We were able to reproducibly quantify 4056 unique proteins (SwissProt identities, protein FDR = 0.01 controlled by Mayu[27]) across all samples. Furthermore, we spiked heavy stable isotope-labeled peptide standards for 20 "anchor proteins" into the T1DS and T2N samples. This approach estimates the absolute abundances of all identified proteins[20]. We additionally measured both twin and unrelated sample sets via the process of dimethyl labeling of

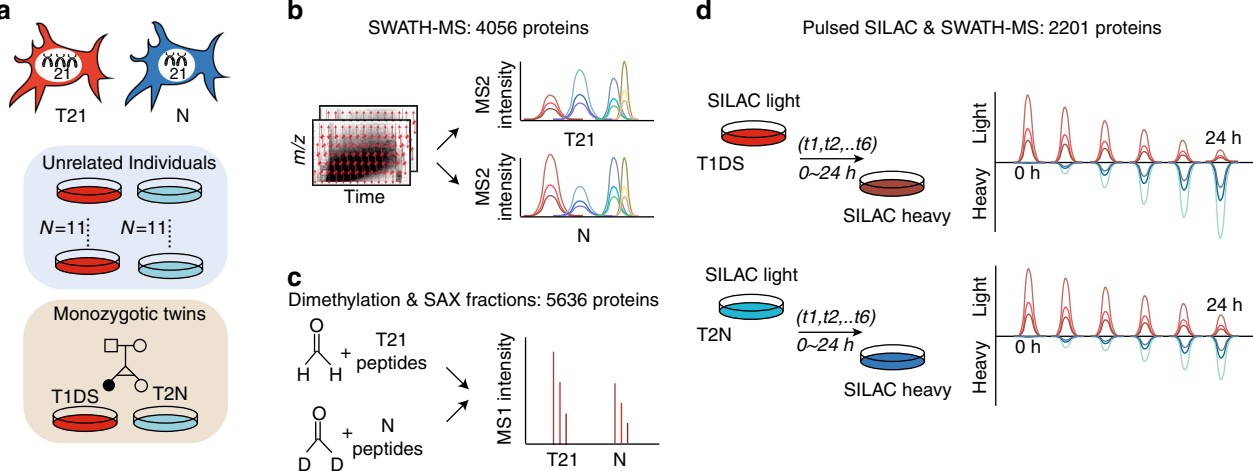

**Fig. 1** Proteome and proteostasis investigation on T21 fibroblast cells. **a** Fibroblast cells collected from a pair of monozygotic (MZ) twins discordant for Trisomy 21 (T21), together with 11 unrelated T21 individuals bearing Down syndrome (DS) and 11 unrelated controls were used for analyzing the effect of T21. T1DS and T2N denote the T21 and normal twin used. **b** A reproducible and accurate data-independent acquisition mass spectrometry (DIA-MS) was used in conjunction with a massively parallel targeted data analysis strategy of SWATH-MS to profile the proteomes. **c** A dimethylation labeling-based shotgun proteomics was used to measure protein abundance change in T21 and N sample mixtures. SAX strong anion exchanger. **d** A pulse SILAC (pSILAC) experiment determines the protein degradation and turnover rates in T1DS and T2N, respectively, across 6 time points over 24 h

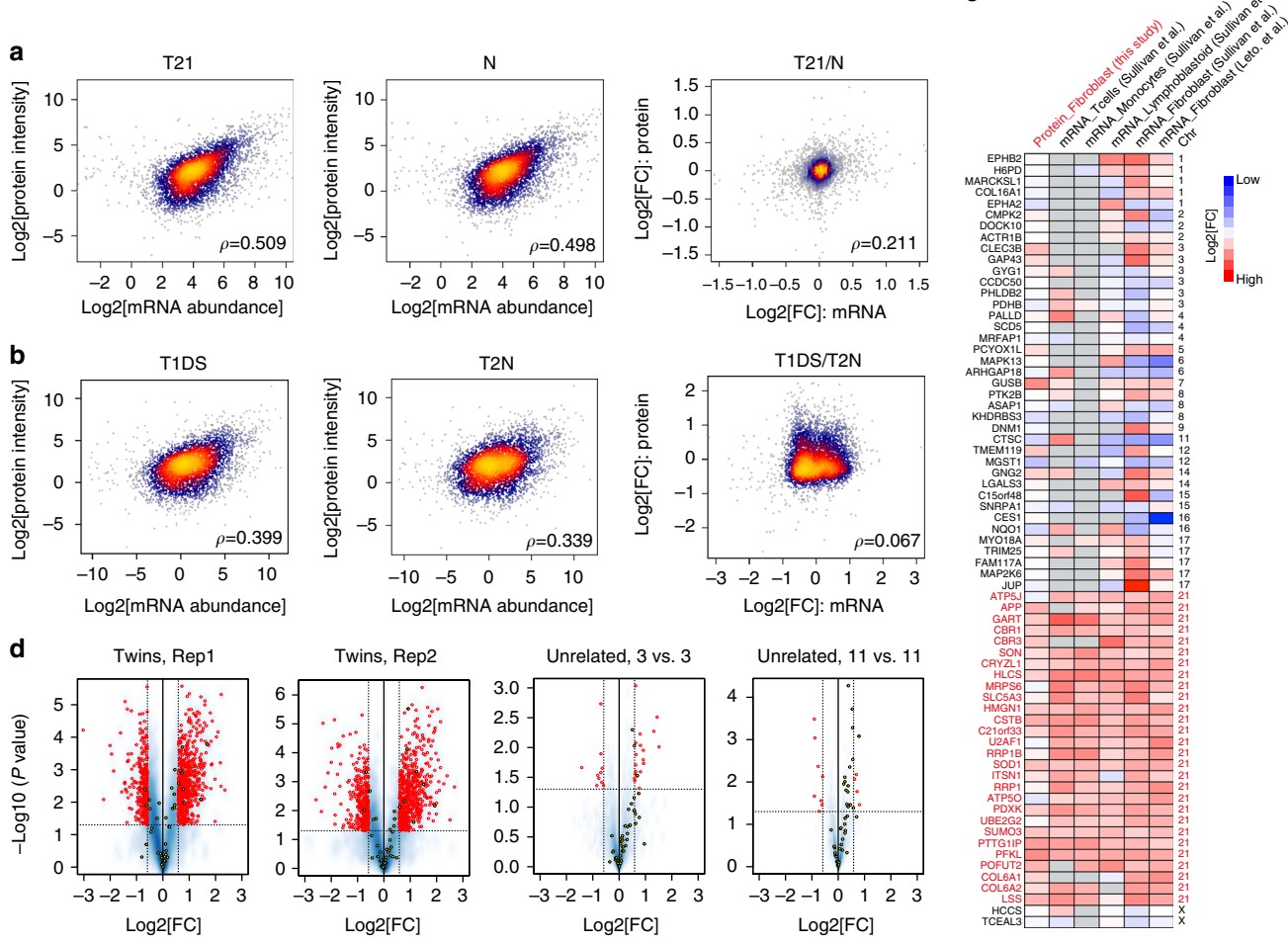

**Fig. 2** Global and significant post-transcriptional regulations revealed by proteomic data. **a** Correlation analysis between mRNA abundance and total protein levels upon T21 revealed a moderate correlation across levels, and weak association regarding cross-layer effects of T21, reflected by matching unrelated samples. **b** Same analysis to **a** but in twin samples. **c** T21 transcript signatures discovered in fibroblast cells of Sullivan et al.[5] are conserved between cell types and impact protein levels. **d** Volcano plots were generated for twin samples (three aliquots of T1DS vs. three aliquots of T2N), three pairs of T21, and N samples (unrelated) and the whole unrelated sample set (11 T21 vs. 11 N). Rep1 and Rep2 mean two independent twin cell cultures. Red circles denote significant proteins of >1.5-fold change (FC) at $P < 0.05$ (Student's t test). The highlighted yellow circles denote Chr21 proteins

sample mixtures, strong anion exchanger (SAX) fractionation, and shotgun proteomics, an alternative proteomic method to corroborate SWATH-MS results (Fig. 1c; Supplementary Figs. 1, 2).

SWATH-MS provided a near-complete data (proteins quantified vs. sample) matrix consisting of more than 4000 proteins across all 56 MS data sets (Supplementary Data 2). This is far superior to existing DS shotgun proteome data sets, in which only ~65.2% of 4023 IPI protein identities (more redundant than SwissProt proteins) were measured consistently in three samples of DS amniocytes[15]. Moreover, compared to the dimethylation proteomics workflow which quantified 5636 proteins in the pooled samples (Supplementary Fig. 2), SWATH-MS of unfractionated samples covered the majority of proteomic events but only consumed one-sixth of the instrument time spent per sample. Importantly SWATH-MS data captured individual variability by excellent technical and biological reproducibility in both absolute and relative protein quantification. An averaged Pearson $R = 0.985$ between all experimental replicates was achieved for both, 22 unrelated samples and T1DS and T2N. And an $R$ of 0.780 for T1DS/T2N ratios of all quantified proteins was achieved between two independent cell cultures (Supplementary Fig. 3). The data further indicate that the dynamic range

of our method covers over five orders of magnitude in protein copy number per cell, ranging from hundreds to tens of millions copies (Methods; Supplementary Data 1).

**The T21 proteome is extensively remodeled.** To assess the global effect of T21 on gene expression at the transcript and protein level, we correlated transcript (from messenger RNA sequencing in Letourneau et al.[14]) and protein concentrations determined in this study. We found moderate correlations ranging between $\rho = 0.339$–$0.508$ in all biological samples, i.e., the twin pair and the nine matched unrelated pairs of DS/N individuals that were also published in Letourneau et al.[14] (Spearman rank correlation $\rho$, Fig. 2a, b; Supplementary Fig. 4). The obtained $\rho$ values are comparable to those reported previously for mammalian cells[9,22,28]. For example, the log-log correlation between the averaged SWATH intensities and mRNA abundances was 0.509 in T21 state and 0.498 in the normal state in unrelated samples. While steady-state transcript and protein abundances showed a moderate level of correlation, transcript and protein DS to N fold change (FC) correlated weakly: $\rho$ was 0.2114 for unrelated individuals and merely 0.067 for T1DS/T2N (Fig. 2a, b; Supplementary Fig. 4c). Similar results were obtained from the

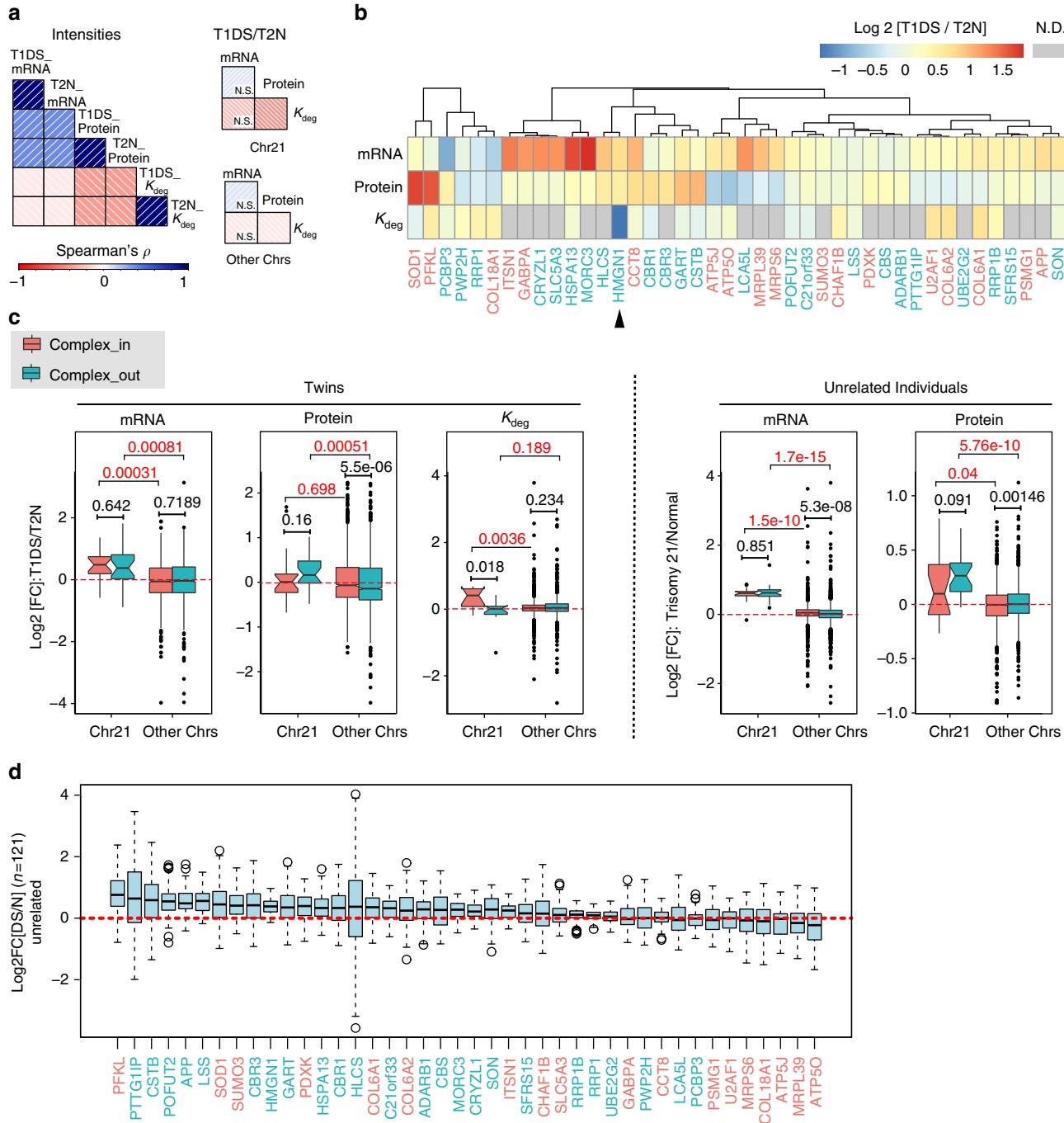

**Fig. 3** Chr21 protein expression and primary dosage compensation. **a** Correlation analysis between absolute and relative changes between mRNA, protein, $K_{deg}$ for global proteome, proteins encoded by Chr21 and other chromosomes. N.S., not significant. **b** Hierarchically clustered heatmap of FC for mRNA, protein, and $K_{deg}$. The red and green colors represent "complex_in" and "complex_out" groups, respectively. N.D., not detected. **c** The dosage compensation against aneuploidy through protein complex stoichiometry control remained apparent at mRNA, protein, and $K_{deg}$ data levels, and is confirmed by the samples from unrelated individuals. $N = 18$ and 24 for complex_in and complex_out groups from Ch21 and $N = 1906$ and 2068 for those from other chromosomes (Other Chrs). **d** Binary log2 fold changes ($N = 121$ comparisons) of Chr21 protein expressions between unrelated individuals. The red and green colors represent "complex_in" and "complex_out" groups. In all box plots (Methods), the borders represent the 25th and 75th quantile, respectively, and the bold bar represents the 50th quantile. Wilcox test was used to infer $P$-values. Standard gene symbols are used, with corresponding SwissProt IDs listed in Supplementary Data 2

dimethylation shotgun data (Supplementary Fig. 4c). The experimental and technical noise in both mRNA and protein measurements may contribute to an underestimation of true correlation. However, as the steady-state abundance values and the calculated FCs were derived from the identical data sets, the result suggests that substantial post-transcriptional regulation accounts for the differential impact of T21 observed on the transcriptome and proteome levels, respectively.

To further investigate the proteomic impact of significant transcriptional events in T21, we overlapped our protein data with T21 transcript signatures in fibroblast cells described by Sullivan et al.[5] Sullivan et al.[5] also analyzed the DS/N

transcriptomes of other cell types, such as lymphoblastoid cells, monocytes, and T cells. We found that the significantly regulated transcripts in fibroblast cells are largely conserved between different cell types, and significantly impact the corresponding 69 protein levels (Fig. 2c). However, about 40% of these 69 proteins are coded on Chr21. Importantly the weak correlations were obtained between the entire transcriptomes of different cell types in Sullivan et al.[5] and again between transcriptomes and our quantitative proteomic data (in general $R < 0.2$, Supplementary Fig. 5). The data suggest that the transcriptional responses for nonChr21 genes caused the general low transcript–protein correlation, which is important to understand the overall T21 phenotype emerged in different cell types.

To evaluate the statistical significance of proteome remodeling, volcano plots were generated for the twin samples (three aliquots of T1DS and T2N, two independent cell cultures), three pairs of unrelated T21 and N samples (randomly selected) and the whole unrelated sample set (11 T21 vs. 11 N). Interestingly, we found a larger number of upregulated proteins than downregulated proteins in T1DS compared to T1N (536–537 upregulated proteins vs. 304–330 downregulated proteins of >1.5 FC at $P < 0.05$ (Student's $t$-test) from all chromosomes (Fig. 2d). In both the 3 vs. 3 and 11 vs. 11 comparisons of unrelated samples, we detected significantly smaller numbers of differentially abundant proteins than the twin samples. We observed that the unrelated genomes significantly buffered the abundance differences of non-ch21 proteins. These results suggest that the absence of genomic variability in the twin samples maximizes our ability to detect small DS/N abundance changes and to thus detect the roots of phenotypic changes. However, the observed patterns strictly only apply to the specific genotype tested. Therefore, our analyses below attempted to interrogate the consistency of regulatory events in a population of genetically unrelated DS/N individuals, in addition to the twins (Supplementary Data 1; Supplementary Fig. 6).

**Gene expression and protein degradation patterns in T21.** In a previous publication in which transcript abundance was compared between T1DS and T2N in the same MZ twin cell system, Letourneau et al.[14] reported gene expression dysregulation domains (GEDDs) organized by chromosome in DS cells. To explore the post-transcriptional effects of the presence of an extra Chr21 in fibroblasts, we used the twin samples and applied pSILAC[21–23] to both T1DS and T2N (three experimental replicates each). Six time points were used for the quantification of protein-specific degradation rates ($K_{deg}$)[21]. With the chase duration over 24 h, about 50% of the total proteome mass was incorporated by heavy lysine and arginine in all cells (Supplementary Fig. 7). The data from the pSILAC experiments quantified the $K_{deg}$ for 2201 proteins (Fig. 1d; Supplementary Data 3).

Comparing the published transcript data of Letourneau et al.[14], the GEDD patterns reported at the transcript level were not apparent at the corresponding protein or protein degradation levels measured in this study (Supplementary Fig. 8). However, we found that protein turnover was generally faster (shorter protein half-lives) in T1DS compared to T2N, with 60% of proteins showing increased $K_{deg}$ rates in T1DS. Furthermore, the absolute T1DS/T2N logFC of $K_{deg}$ was on average much smaller than for mRNA or protein levels (Supplementary Fig. 9). Only 11.2% of the measured degradation rates have FCs above 1.3, compared to 46.9% of the protein levels and 54.5% of the mRNA levels (Supplementary Fig. 8). The stability of $K_{deg}$ could be partially explained by the absolute protein abundance per cell and the haploinsufficiency of the gene, i.e., highly abundant and haploinsufficient proteins tend to be less strongly regulated by

protein degradation rates than other proteins, as expected[29] (Supplementary Figs. 10, 11). Altogether, the proteome dynamics upon T21 demonstrates a similar extent of dysregulation as compared to transcriptional changes, but seems not to be organized through the chromosome domains reported at the transcript level[14].

**Effects of T21 on Chr21 proteins.** We focused on the overall expression of Chr21 genes, since they are likely to lead the development of DS phenotypes (Supplementary Fig. 12). In total, 42 Chr21 proteins were quantified by SWATH-MS and 56 by dimethylation shotgun proteomics, representing 18.7–24.9% of the 228 protein coding genes on Chr21, respectively[14]. In the twin samples, we found on average FCs for mRNAs, proteins and $K_{deg}$ of 1.38, 1.13, and 1.07, respectively, for Chr21 genes (Supplementary Fig. 8). The correlation between $K_{deg}$ and protein FC of T1DS/T2N was weakly negative for all proteins across the genome ($\rho = -0.051$), but was strongly negative for proteins coded on Chr21 ($\rho = -0.416$). The significant difference between the two correlations ($P = 0.04$, $z$-test) indicates the importance of protein degradation in shaping Chr21 gene expression in T21 (Fig. 3a).

Next, we analyzed individual Chr21 genes (Fig. 3b). First, we found that 18 out of 42 mRNAs and 9 out of 42 proteins displayed a FC > 1.4 in T21[2]. Among these, four genes (GART, CCT8, HLCS, and HMGN1) were dysregulated both at the mRNA and protein levels, indicating that they are Chr21 dosage-sensitive. The upregulation of HMGN1, previously detected at the transcript level, has been hypothesized to increase chromatin accessibility and cause transcriptional dysregulation[14]. The degradation of HMGN1 in T21 was significantly reduced (by almost 2.5 FC) in our data, supporting the hypothesized need of upregulating HMGN1 expression in trisomic state[14]. SOD1, PFKL, and CSTB proteins showed the largest concentration increase in T21. They were upregulated by 3.21, 3.01, and 2.02-fold, respectively. Surprisingly, their corresponding transcripts were not substantially changed (1.15, 0.96, 1.13 FC, respectively), indicating a significant post-transcriptional upregulation for these proteins. We also observed the opposite pattern: some proteins encoded on Chr21 were buffered, i.e., their protein levels remained close to unchanged, despite substantial upregulation of their coding transcripts. For example, *U2AF1*, *COL6A1*, and *COL6A2* transcript expression increased by 1.18, 1.40, and 1.18-fold in T1DS/T2N, but were buffered to FC values of 0.87, 0.91, and 0.85, respectively, at the protein level. Consistently these three proteins showed the strongest increase in protein turnover among Chr21-coded ones. Specifically, $K_{deg}$ increased by 1.57, 1.60, and 1.59 FC for U2AF1, COL6A1, and COL6A2 in T21. Hence, the buffering of protein levels most likely resulted from adapted protein turnover rather than reduced protein synthesis rates[9,11]. To sum up, in response to T21 we discovered non-uniform expression patterns for Chr21 genes suggesting complex and varying modes of regulation.

The tight control of protein complex stoichiometry has been reported as a buffering mechanism against aneuploidy in different systems[10,11,30,31]. To analyze the contribution of protein complex stoichiometry to the T21 phenotype, we classified proteins into two groups depending on their inclusion ("complex_in") or exclusion ("complex_out"; Methods) from known stable heteromeric protein complexes[32,33], and compared their log2 FCs. Transcripts encoding for either, "complex_in" or "complex_out" proteins from Chr21 ($N = 18$ and 24, respectively) had similar, positive FCs (Fig. 3c), which were significantly higher than mRNAs transcribed from other chromosomes ($P = 0.00031$ and $P = 0.00081$, Wilcoxon test—the same test used below). Despite the similar mRNA level changes, we observed important

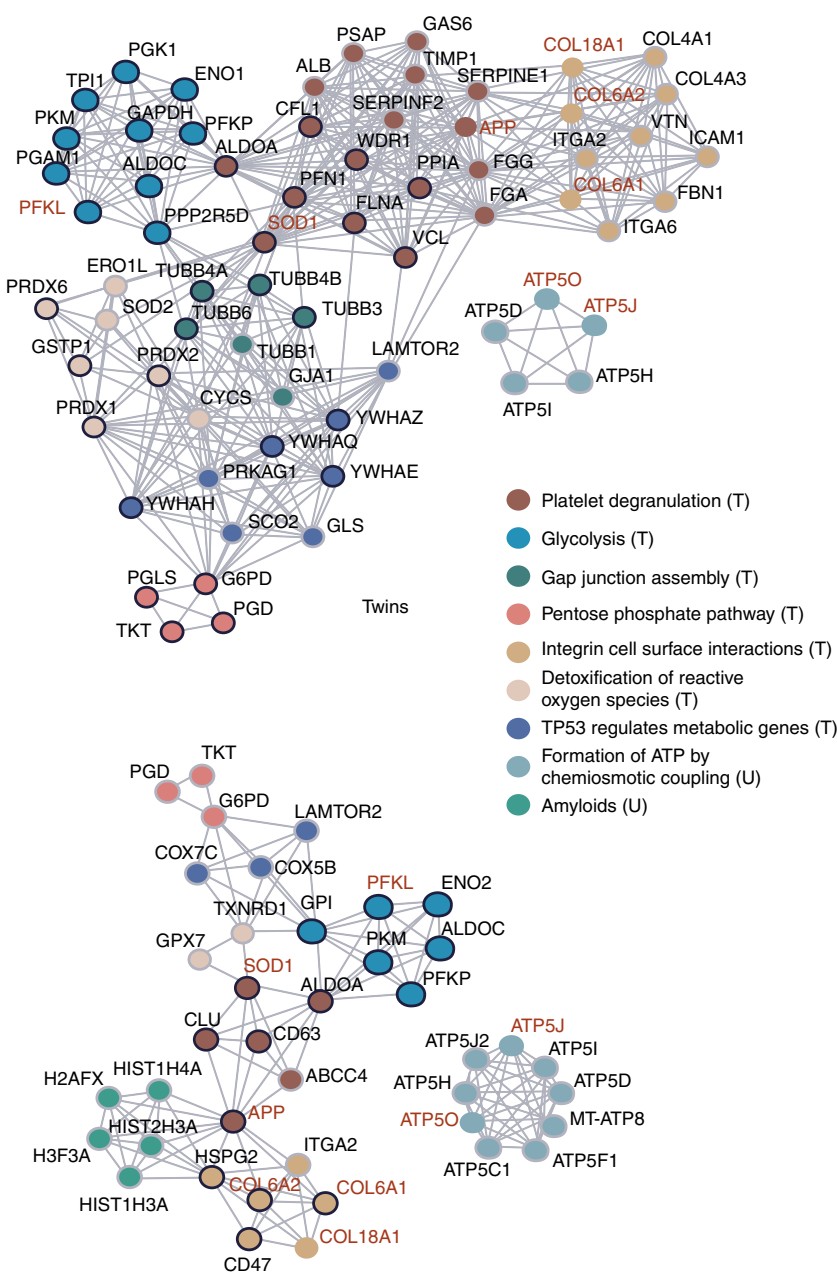

**Fig. 4** Significantly regulated proteins and their network associated to T21. Proteins in the reactome pathways that were found to be significantly enriched ($P < 0.01$, see Methods for details) are represented as a network graph, where edges indicate functional connections, i.e., links between proteins in same reactome pathways. Nodes represent individual proteins and these are colored according to a dominant reactome pathway annotation. If the protein is upregulated, its node border is black, if it is downregulated it is light gray. Names of the proteins encoded by the genes on the chromosome 21 are shown in red

differences between "complex_in" and "complex_out" Chr21 proteins: protein FCs of the "complex_in" group were indistinguishable from "complex_in" proteins on other chromosomes ($P = 0.698$). In contrast, the expression of "complex_out" Chr21 proteins in T1DS was significantly upregulated compared to other chromosomes ($P = 0.00051$). Identical observations were made in the 22 unrelated samples (Fig. 3c) and were confirmed by independent dimethylation shotgun proteomics analyzing both twin and unrelated samples (Supplementary Fig. 13). Our observations were further confirmed by the protein turnover measurements: the $K_{deg}$ of "complex_in" Chr21 proteins in T1DS was significantly higher than both "complex_out" Chr21 proteins

($P = 0.018$) and "complex_in" proteins on other chromosomes ($P = 0.0036$). *U2AF1*, *COL6A1*, and *COL6A2* are typical gene examples encoding such "complex_in" Chr21 proteins that are strongly buffered.

Buffering of protein levels against mRNA variation can occur at multiple scales, including intra- and inter-individual genomic variation[9,34]. Therefore, we examined the inter-individual variability within the unrelated samples, by calculating the T21/T2N FC for all proteins between all the possible sample pairs using unrelated samples (i.e., $11 \times 11 = 121$ pairs) and determined the variability of the protein FC. The 42 Chr21 proteins quantified in this study demonstrated varied FC and

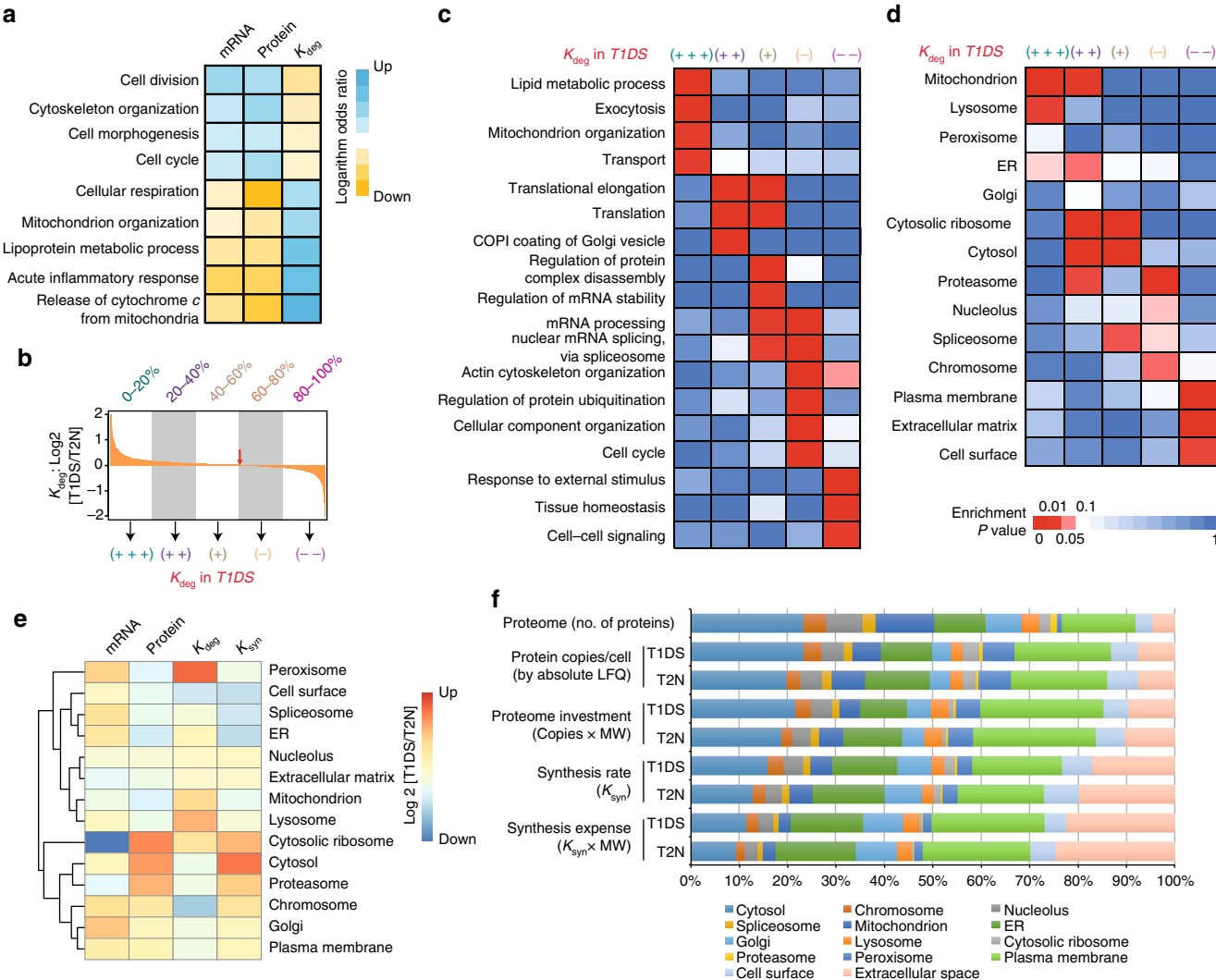

**Fig. 5** Extent of protein degradation reveals organelle-related gradients in T21. **a** Selected significant, overlapping biological processes after GESA analysis at mRNA, protein, and $K_{deg}$ levels. The logarithm odds ratios (LOR) were visualized on a red–blue color scale, with blue color denoting LOR > 0 (i.e., T1DS gene products are under represented) and red denoting LOR < 0 (i.e., T1DS products are over-represented). **b** The five classes of proteins based on $K_{deg}$ regulation extent. The red arrow denotes zero $K_{deg}$ regulation. **c** The ranked list of protein degradation regulation under T21 stress was divided into five segments of proteins for which GO biological processes enriched in each segment were selected and displayed. **d** Cellular components annotation and enrichment analysis indicates the extent to which protein degradation forms an organelle-related gradient in T21. **e** The transcript, protein, protein degradation, and protein synthesis FC of all the proteins annotated in each organelle were averaged for hierarchical clustering analysis. **f** Comparing the number of proteins, absolute protein copies, protein investment, and synthesis expense for different organelles in T1DS and T2N, respectively. $K_{deg}$ protein degradation rate, $K_{syn}$ synthesis rate

quantitative variability among individuals, with HLCS being the most variable (Fig. 3d). The upregulation of "complex_in" proteins tended to be less variable among DS individuals than those of "complex_out" proteins ($P = 0.0307$, one-tailed $\chi^2$-test based on half-half list splitting), confirming the buffering through protein complexes discovered from the twin samples. Collectively, our data indicate that the coordinated protein degradation within protein complexes is largely responsible for the primary buffering against excessive transcript and protein dosages due to T21, which holds true not only within one individual cellular system, but also between genetically diverse individuals. These important regulatory effects were not apparent from the transcript data alone.

Finally, in the twin samples, the $K_{deg}$ for both "complex_in" and "complex_out" proteins encoded on chromosomes other than Ch21 did not show a difference ($P = 0.234$, Fig. 3c). Thus, whereas protein degradation counteracts the gene dosage effect

for proteins encoded on Chr21—especially for members of protein complexes—we observed concordant regulation of transcription and protein turnover for proteins encoded on other chromosomes (see also next sections).

**Organelle-specific protein expression effects of T21.** To reveal the proteomic impact of T21 on cellular functions, we used reactome pathways[33] to connect Chr21-encoded proteins to the 406 proteins with the highest 5% and lowest 5% T1DS/T2N FCs (Fig. 4). The resulting network significantly enriched several reactome pathways (Supplementary Data 4), such as glycolysis ($P = 2.10e-05$; Methods), platelet degranulation ($P = 5.44e-06$), integrin cell surface interaction ($P = 0.0083$), and pentose phosphate pathway ($P = 0.0044$). Strongly overlapping sets of pathways were observed as enriched when protein data from the 22 unrelated samples were used (here we selected 406 most strongly

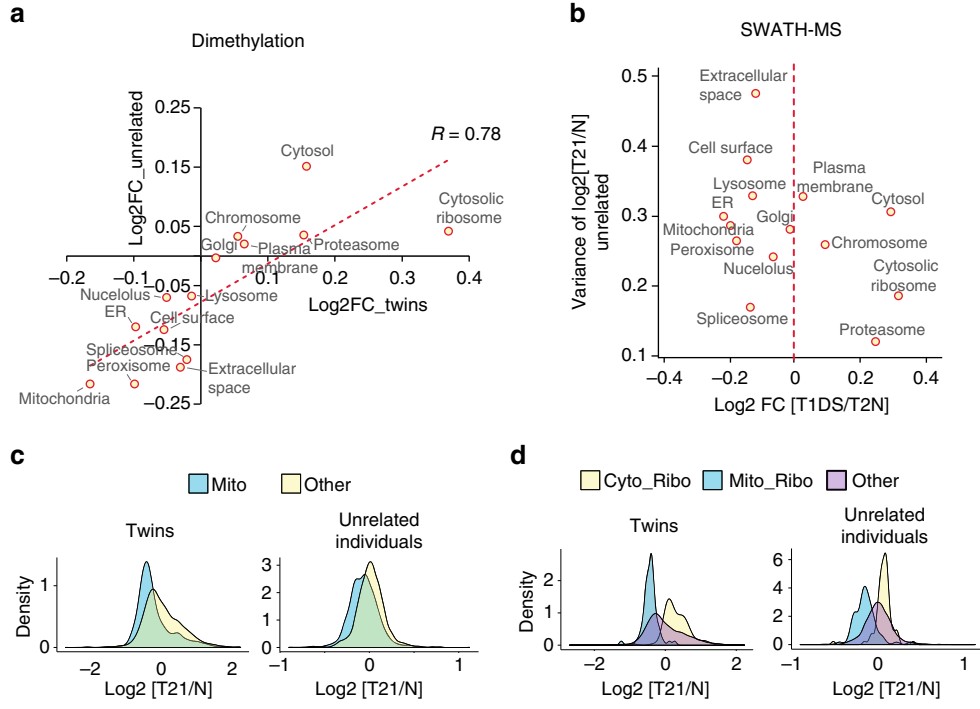

**Fig. 6** Inter-individual variability of organelle proteome alternations induced by T21. **a** High correlation of organelle proteome regulation trends between twins and unrelated samples. **b** The correlation between $\log_2$ T1DS/T2N ratios for each organelle and the inter-individual variability of T21/normal FC using unrelated samples. ER, endoplasmic reticulum. **c** The distribution of T21/normal ratios of mitochondrial proteins and other proteins in the proteome, for twin samples and unrelated individuals. **d** The distribution of T21/normal ratios of cytosolic ribosome proteins (Cyto_Ribo), mitochondrial ribosome proteins (Mito_Ribo), and other proteins in the proteome, for twin samples and unrelated individuals

regulated proteins whose inter-individual CVs were <50% within the normal group), although the observed FC values were attenuated. The interferon alpha–beta signaling, which was identified as a T21 transcriptional signature[5], has a marginal $P$-value of 0.056 at proteome level, perhaps due to the smaller genome coverage by proteomics. These findings suggest that common cellular functions are affected by T21 at the proteome level across individuals, such as energy metabolism and cell surface signaling processes.

To extend the analysis of T21 perturbed pathways to the transcript[14] and protein degradation level, we performed gene set enrichment analysis (GSEA) to identify biological processes that are consistently associated to high or low FCs at each layer[35] (Supplementary Data 5). The intersecting processes between the protein, transcript, and protein degradation layers denoted biological functions that were significantly and commonly altered at all three layers. These included cell cycle-related functions, cell morphogenesis, lipoprotein metabolism, and cellular respiration in mitochondria (adjusted $P < 0.05$ at all layers, Fig. 5a; Supplementary Fig. 14). There is a remarkably reciprocal regulatory pattern between mRNA and protein turnover. For example, proteins related to the cell cycle and morphogenesis are transcriptionally induced in T21 and have lower $K_{deg}$, whereas proteins responsible for cellular respiration process are transcriptionally tuned down and have higher $K_{deg}$. This reciprocal pattern suggests a concordant regulation of these protein levels through synthesis and turnover (Fig. 5a).

To systematically characterize proteome-wide protein degradation in T21, we divided the protein list based on $K_{deg}$ FC of T1DS/T2N into five even sets (quintiles), and asked which GO biological processes were enriched in each quintile (Fig. 5b). For example the quintile with the highest $K_{deg}$ in T1DS (labeled as +++) was enriched for lipid metabolic process ($P = 0.00136$), exocytosis ($P = 0.0037$), mitochondrion organization ($P = 0.0085$), and

protein transport ($P = 0.0099$) (Fig. 5c; Supplementary Data 6). Similarly distinctive patterns were observed for other quintiles. These enriched GO biological processes significantly associate to different cellular components/compartments. We therefore performed the same enrichment analysis based on GO cellular component annotations. Different organelles and cellular components are strikingly, and in most cases uniquely, enriched based on the extent of protein degradation regulation in T21 (Fig. 5d; Supplementary Data 7). Specifically, those genes that have the highest $K_{deg}$ rates in T1DS are drastically enriched for mitochondrial genes (+++: $P = 1.52e{-}8$; ++: $P = 0.0074$). Lysosome genes were uniquely enriched among the ones with the highest $K_{deg}$ (+++; $P = 0.014$), while cytosolic ribosome genes have slightly elevated $K_{deg}$ (++; $P = 4.60e{-}17$; +; $P = 5.18e{-}15$). Other gene categories had relatively unchanged $K_{deg}$, such as cytosol (+; $P = 1.36e{-}7$), spliceosome (+; $P = 0.0267$), proteasome complex (−; $P = 1.58e{-}4$), and chromosome (−; $P = 0.036$). Remarkably, the turnover of plasma membrane (−−; $P = 3.02e{-}4$) and cell surface (−−; $P = 0.0166$) related proteins were greatly slowed down in T21. In summary, our data strongly indicate that an extra copy of Chr21 triggers an organelle-dependent and coarse-grained protein degradation gradient in fibroblasts.

Next we explored the organelle-specific gene expression control in T21 (Fig. 5e). We estimated steady-state protein synthesis rates ($K_{syn}$) (Methods). First, we found that $K_{syn}$ globally recapitulates the protein level regulation. Thus, protein degradation generally fine-tunes protein levels after synthesis in most organelles. Second, we discovered that in mitochondria, peroxisomes, and lysosomes, protein degradation prominently buffered against mRNA variation to shape the protein levels. The dimethylation proteomic data showed consistent patterns of organelle regulation at the protein level (Supplementary Fig. 15a). Third, T21 turned out to have a limited effect on nucleolar gene expression. Fourth,

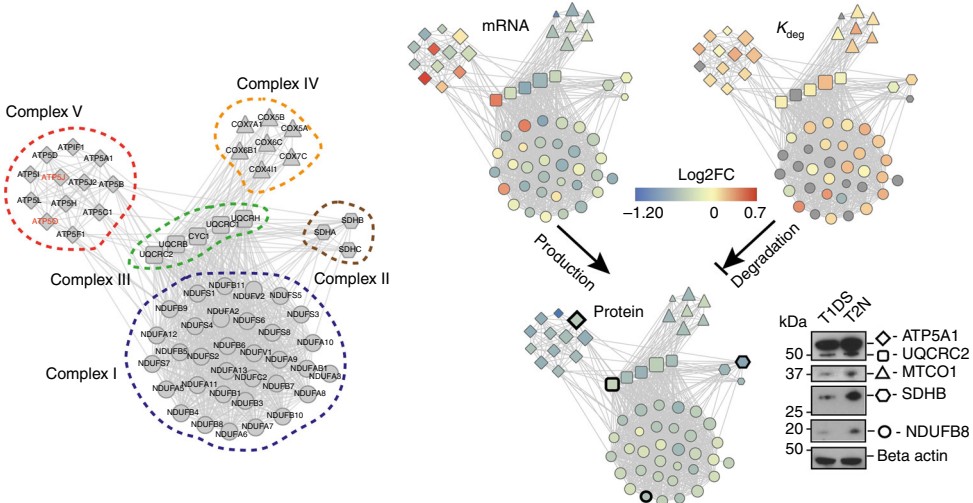

**Fig. 7** Co-regulation of mitochondrial protein complexes in T21. Co-regulation of mitochondrial complexes starts at the transcriptional level and gets uniform at protein level through protein level regulation and protein degradation processes. Proteins belonging to Complexes I–V are visualized by the nodes with shapes of circle, hexagon, rectangle, triangle, and diamond, respectively. The edges between nodes denote high-confidence protein interactions documented in STRING database (minimal combined score >0.45). In the visualization panels of mRNA, protein, and $K_{deg}$ comparisons, the node size is proportional to the $\log_{10}$ scale transformed protein copies. The color gradient of blue–yellow–red depicts the $\log_2$ T1DS/T2N FC values. Standard gene symbols are used, with corresponding Swissprot IDs listed in Supplementary Data 2. The Western blots were performed for selected proteins in each mitochondrial protein complex, confirming their downregulations in T1DS

although ribosome-related transcripts were suppressed and ribosomal protein degradation accelerated, the cytosolic ribosomal proteins were clearly upregulated in T21 cells, strongly suggesting that the translation rate of ribosome-related genes was massively increased, an observation for which the transcriptomic data alone may lead to contradictory conclusion[5]. Finally, we used a recently published subcellular map of the human proteome, derived from antibody-based microscopy experiments, as an alternative annotation system of organelles[36], and obtained identical patterns as the results showed in Fig. 5d (Supplementary Fig. 16). Altogether, our multilayered data essentially dissected gene regulation in each organelle under the aneuploidic stress of T21.

Our integrative data of T1DS and T2N permits systems-level accounting of absolute proteomic cost for cells and relative cost when they bear an extra Chr21. We therefore calculated the proteome investment (copies per cell times protein molecular weight (MW)) and synthesis expenses for different organelles (Fig. 5f). We found that cytosolic proteins, despite comprising ~23% of the total quantified proteome, consumed only about 10% of the total synthesis expense in T1DS and T2N cells, indicating that these proteins have a generally slow turnover. In contrast, the peroxisome constitutes only 0.88% of the quantified proteome, but these proteins represent 6.6% of the absolute protein copies in T1DS and T2N samples. Meanwhile, their synthesis expense is of much lower percentage (~1.75%), indicating slow $K_{deg}$. The opposite pattern is observed for the Golgi-related proteins. Interestingly, when T2N and T1DS are compared, we found that protein investment and synthesis rate and expense are all smaller in T21 cells for proteins of the extracellular space and plasma membrane. Again, the opposite pattern is observed for cytosolic proteins and ribosome proteins, suggesting that T21 cells seem to invest more on inside the cells than on the cell surface.

The manifestation of Chr21-induced organelle aberrations discovered above from twin samples is modulated by genomic differences and/or environmental factors in the genetically different DS cases tested. Whereas the overall correlation of T21/T2N FC for all the quantified proteins between twin and unrelated sample sets is modest ($R = 0.200–0.279$, Supplementary

Fig. 15b, c), we observed a significant consistency of organelle proteome aberrations between twins and unrelated samples, when the individual variations are averaged by sample mixing and dimethylation based quantification ($R = 0.78$; Fig. 6a). Moreover, we found that the absolute FCs of organelle proteomes were typically smaller in the unrelated individuals compared to the twins using individual SWATH-MS data sets (Fig. 6b–d). Specific cellular components such as lysosome and cell surface proteins showed high variability in regulatory trends between unrelated DS individuals (Fig. 6; Supplementary Fig. 17). Thus, we deduce those organelle-based functions that are consistently regulated between unrelated individuals, such as cell cycle regulation, ribosome translation, and mitochondrial respiration (Fig. 5a), likely drive several cellular hallmarks of DS phenotypes. Conversely, the most variable organelles, such as lysosome exocytosis and cell–cell surface signaling, may contribute to the variability of DS phenotypes.

**Co-regulation of mitochondrial protein complexes.** Mitochondrial downregulation emerges as one of the most significant DS phenotypes, which was confirmed in both discordant twins and unrelated individuals (Fig. 6c). Specifically, we quantified 33, 3, 5, 7, and 12 proteins from the mitochondrial respiratory complexes I–V, respectively, and among these we quantified the $K_{deg}$ changes for 22, 2, 4, 6, and 11 proteins in T1DS and T2N. For these genes, 49 out of 60 (81.6%) transcripts are downregulated in T21. Remarkably, all of 60 genes (100%) are downregulated at the protein level in T1DS with an average FC of 1.39, which can be partially ascribed to the fact that 84% of these proteins increased their $K_{deg}$ by an average of 11.2% (Fig. 7). Also, 53 out of 60 proteins (88.3%) are downregulated in unrelated DS based on their averaged FC. The uniform downregulation of mitochondrial respiration complexes was verified by Western blotting analysis (Fig. 7; Supplementary Fig. 18). Among these complexes, complex V (ATP synthase) is of particular interest, because it includes two identified Chr21 proteins, ATP5J and ATP5O. Although we observed increased transcript levels of *ATP5J* and *ATP5O* due to T21, protein levels were clearly below normal. Altogether, our data provide evidence for a strong repression of

respiratory chain complexes due to T21, which is consistent with prior work[37,38].

## Discussion

Due to the previous technical limitations of proteomics, the impact of Chr21 gene expression on DS phenotypes thus far has been based on (i) the established functions of Chr21 proteins from individual biochemical studies, and (ii) the dosage measured at the transcript level in DS samples[2,39]. Such studies may, however, be inadequate due to the existence and complexity of post-transcriptional regulation in mammalian systems[9,22,23]. To understand molecular mechanisms perturbed by T21 in more depth to potentially identify actionable therapeutic targets, it is essential to probe the gene dosage imbalance in DS at the protein level. Indeed, we found that the correlation between the previously reported mRNA and our protein FC upon the gain of the extra Chr21 was weak (Fig. 2a, b), demonstrating the importance of studying the entire proteome. To date, proteome-wide abundance and turnover measurements have been lacking for T21 cells, indicating our data set itself a significance resource for DS research. For example, the dosage-sensitive proteins escaping the protein level buffering mechanisms discovered by our study could be more relevant to DS development and phenotypes. Compared to previous studies, which only measured a few proteins in clinical DS samples[16,17], in the present study several state-of-the-art proteomic methods including SWATH-MS, absolute label-free quantification (aLFQ), and pSILAC were jointly applied. SWATH-MS quantification is based on multiple fragment ion chromatograms per quantified peptide and has been shown to provide selectivity and accuracy comparable to selected reaction monitoring (SRM) measurement[18,19], the gold standard quantitative mass spectrometric method. Further, SWATH-MS has shown lower limit of detection than MS1-based profiling under comparable experimental conditions[18,40]. Our SWATH-MS-based label-free quantification achieved sufficient accuracy and desired statistical significance in this study. Collectively, the techniques used have broken through a major technical hurdle of studying the effects of T21 on the proteome.

In this study, we used fibroblast cells for reasons of easy accessibility as proxies to understand the impact of T21 on basic cellular processes in DS. There are at least three phenotypes of DS potentially reflected by fibroblast cells (See Supplementary Table 1 for literature summary): (a) endosome pathology related to Alzheimer;[4,41] (b) apoptotic phenotype;[42] (c) Higher oxidative stress related to premature ageing. Recently, the transcriptional activation of the interferon response pathway due to T21 was also first discovered in fibroblast[5]. Nevertheless, the use of fibroblast may limit the impact of certain findings present and the study of other cell types and tissues will be necessary to fully understand the specific effects and other phenotypes of T21.

In DS, the observed (molecular) phenotypes are the result of genomic variability between individuals and the third copy of Chr21. The combination of the twins and unrelated samples provides a unique perspective for understanding the specific consequences of T21, and permits the calculation of inter-individual variability both exclusively within Chr21 and across the rest of the genome. It was reported that frequently genetic variance affecting protein abundance has no effect on mRNA levels[34,43]. Our study confirmed this extra dimension of complexity at the proteomic level. HLCS, a Chr21 protein that participates in chromatin condensation and gene repression[14,44], was dosage-sensitive in both twin and unrelated samples, but showed highest quantitative variability among DS individuals (Fig. 4a). Other variable consequences of T21 are largely caused by genomic variation outside Chr21. For example, for proteins of the

class "cell-cell surface signaling" we found considerable variability between unrelated individuals, suggesting that their abundance is affected by genomic and environmental variation. Additionally, proteins in the lysosome and endoplasmic reticulum groups also exhibit large inter-individual variation, likely contributing to the biological variation between DS individuals.

We observed a variable, organelle-dependent protein degradation extent resulting from T21. Previously the cellular compartment-specific differences in $K_{deg}$ had been observed in a steady-state system by blocking the proteasome[45]. Also, the mitochondrial and cell surface signaling alterations have been previously observed at the transcript level in DS[46]. Whether these organellar abberrations (Fig. 3) are specific for T21 or occur in response to any genome gain aneuploidy remains to be discovered.

Previous studies have investigated the proteomes of different aneuploidy systems in yeast[11], fly[30], human cancer cells[10], and in artificial microcell fusion-induced human cells carrying different additional chromosomes[7,31]. Here we quantified both protein and protein degradation in a unique cohort of T21 primary cells to study the impact of T21 aneuploidy. Our data indicate a mild but global acceleration of protein turnover in T21 cells. Further, we found that those Chr21 proteins that are members of heteromeric protein complexes were less affected by the increased gene dosage due to aneuploidy, a buffering effect that is mediated by accelerated protein degradation. Although tight stoichiometric control of protein complex subunits was previously reported as a buffering mechanism in aneuploidy[10,11,30,31], no studies have yet directly quantified the regulation of Chr21-wide protein turnover or in clinically derived samples. It has been a long-standing hypothesis that proteins are stabilized by complex formation and by protein degradation against misfolded proteins[47]. The concept has been investigated with many individual studies as reviewed recently[9]. We argue that by generating sample-specific protein complex data sets (e.g., by size-exclusion chromatography[48]), our finding can be further extended and its significance generalized. Furthermore, previous studies have debated the extent (or even existence) of dosage compensation in eukaryotic cells[49,50]. Our data show that—at least in T21—protein dosage compensation exists and plays a primary role in the mechanistic processes underlying DS.

Altogether, our data distinguish two classes of proteins. The first class consists of proteins for which it is apparently necessary for the cell to regulate the abundance in order to cope with the stress induced by the extra Chr21. Examples of this class are instances with consistently upregulated mRNA and protein levels at reduced $K_{deg}$, as well as instances with consistently reduced protein expression (Fig. 5a). The second class contains cases for which altered mRNA abundance levels are buffered at the protein level, i.e., where the protein FC is smaller than RNA FC. In some cases this is achieved by an increased $K_{deg}$. We discovered that the second class is particularly prominent on Chr21 (Figs. 2c, 3c), because the cell cannot change the copy number state. Responses of proteins on other chromosomes might be a mix of both classes.

Finally, our data clearly demonstrate that the mitochondrial proteins in the five ETC complexes are significantly and uniformly downregulated in T21. Using astrocytes and neuronal cultures from DS fetuses, Busciglio et al.[51] showed in 2002 that altered metabolism of the amyloid precursor protein (APP) and oxidative stress result from mitochondrial dysfunction. The transcriptional and functional analyses have showed that the reduced mitochondrial activity in DS acts as an adaptive response for avoiding injury and preserving basic cellular functions in DS[37]. Moreover, the mitochondrial alteration affecting reactive oxygen species homeostasis was suggested to lead the pathogenesis of various neurodevelopmental syndromes including DS[38].

Recently genes implicated in energy metabolism, mitochondrial activity, and biogenesis are reported to be downregulated in the Ts3Yah mice, which carry a duplication syntenic to human 21q11.2-q21.3[52]. In our results, the dysfunctional mitochondria were apparent at both protein and protein degradation levels, significantly reinforcing the previous observations. Such knowledge may pave novel avenues for DS therapeutics, e.g., by activating mitochondrial function, reducing oxidative stress, and thus ameliorating the physiological DS symptoms. As promising examples, novel translational approaches involving the utilization of coenzyme Q10 (CoQ10) were used to treat a variety of clinical phenotypes associated with DS[53]. Also, the use of metformin was recently found to be effective to reverse mitochondrial dysfunction in DS cells, thus exerting protective effects against DS-associated pathologies[54].

In summary, we expect that this proteome and proteostasis profiling data set illustrates the significance of proteome level dosage analysis for genetic diseases like human DS and will benefit future studies.

## Methods

**Samples and cell culture.** The forearm primary fetal skin fibroblasts were collected post mortem from the T1DS and T2N discordant twins at 16 fetal weeks, after IRB approval from the Ethical Committee of University Hospitals of Geneva, and written informed consent by both parents[14]. Two vials of cells were thawed and cultured separately; the two cultures represent the two biological replicates. Next, biological replicates were divided into three cultures and handled as technical replicates. To compare and generalize our finding, we also analyzed two technical replicates of unrelated primary skin fibroblasts from 11 DS individuals and 11 unaffected individuals (Supplementary Data 1). The written informed consent was obtained from all human participants in this study[4]. All primary fibroblasts were grown in DMEM with GlutaMAX (Invitrogen, Life Technologies) supplemented with 10% fetal bovine serum (Life Technologies) and 1% penicillin/streptomycin/fungizone mix (Amimed, BioConcept) in a 37 °C incubator supplied with 5% $CO_2$.

**Pulsed SILAC experiment.** For the pSILAC experiment, SILAC DMEM High Glucose Medium (GE Healthcare) lacking L-arginine, L-lysine was supplemented with either light or heavy isotopically labeled lysine and arginine, 10% dialyzed fetal bovine serum (PAN Biotech), and 1% penicillin/streptomycin mix (Gibco). Specifically, 146 mg per liter of heavy L-lysine ($^{13}C_6$ $^{15}N_2$) and 84 mg per liter of arginine ($^{13}C_6$ $^{15}N_4$) (Chemie Brunschwig AG) and the same amount of corresponding non-labeled amino acids (Sigma-Aldrich)[22] were supplemented, respectively, to configure heavy and light SILAC medium. Additionally 400 mg per liter L-proline (Sigma-Aldrich) were also added into SILAC medium to prevent the potential arginine–proline conversion. The three replicates of T1DS and T2N samples were used. The identical fibroblast cell aliquots were first split and grown in the light SILAC medium. At time point zero ($t_0 = 0$ h), the medium were switched into heavy SILAC medium upon the gentle washing of three times by pre-warmed PBS, and totally six time points were included (0, 1, 4, 8, 14, and 24 h) to configure the pSILAC course. At each time point, the cells were collected as snap frozen pellets after three times of pre-cold PBS washing. The cell pellets were stored in −80 °C for proteomic analysis.

**Cell doubling time determination.** The cell doubling time T1DS and T2N was carefully conducted by triplicate cell counting at nine time points (0, 7, 24, 31, 48, 55, 72, 79, and 96 h) on a hemocytometer. Three independent experiments were conducted in three technical replicates. The average doubling times for T1DS and T2N were determined to be 48.4 and 43.3 h, respectively ($P = 0.049$, Student's $t$ test). The formula used to calculate the doubling time is: Duration × log(2)/log (FCN)–log(ICN) where Duration = duration of the cell growth, FCN = the final number of cells at end of a given time point, ICN = the initial number of cells at time point 0 h, i.e., inoculum.

**Protein extraction and in-solution digestion.** The cell pellets were suspended in 10 M Urea lysis buffer and complete protease inhibitor cocktail (Roche), ultrasonically lysed by sonication at 4 °C for 2 min using a VialTweeter device (Hielscher-Ultrasound Technology), and then centrifuged at 18,000×$g$ for 1 h to remove the insoluble material. The supernatant protein mixtures were reduced by 10 mM tris-(2-carboxyethyl)-phosphine (TCEP) for 1 h at 37 °C and 20 mM iodoacetamide (IAA) in the dark for 45 min at room temperature. All the samples were further diluted by 1:6 (v/v) with 100 mM $NH_4HCO_3$ and digested with sequencing-grade porcine trypsin (Promega) at a protease/protein ratio of 1:25 overnight at 37 °C. The amount of the purified peptides was determined using Nanodrop ND-1000 (Thermo Scientific) and 1 μg peptides was injected in each LC-MS run.

**Shotgun proteomics on triple TOF.** The peptides digested from cell lysate derived from normal cell culture and lysate and the samples of the pSILAC experiment were measured on an SCIEX 5600 TripleTOF mass spectrometer operated in DDA mode[18]. An Eksigent NanoLC Ultra 2D Plus HPLC system and a 20 cm PicoFrit emitter (New Objective, self-packed to 20 cm with Magic C18 AQ 3 μm 200 Å material) were used for peptide separation. For the HPLC method, the buffer A used was 0.1% (v/v) formic acid, 2% (v/v) acetonitrile, whereas the buffer B was 0.1% (v/v) formic acid, 98% (v/v) acetonitrile. We adopted a 120 min gradient of 2–35% buffer B with a flow rate of 300 nL/min. For MS method, a survey scan at the MS1 level (360–1460 $m/z$) was first carried out with 500 ms per scan. Then, the shotgun settings were designed to sequence the Top20 most intense precursors, whose charge states are 2–5. Signals exceeding 250 counts per second were selected for fragmentation and MS2 spectra generation. MS2 spectra were collected in the mass range 50–2000 $m/z$ for 100 ms per scan. The dynamic exclusion parameters were set so that already sequenced precursor ions were excluded from reselection for 20 s duration.

**Shotgun proteomics on Orbitrap Elite.** The pSILAC samples were also measured on Oribtrap Elite (Thermo Scientific) for comparison between the conventional method and the SWATH-MS-pSILAC pipeline. Briefly, 1 μg of peptide sample was injected per run and a Thermo Easy-nLC 1000 HPLC system with a 15 cm long PepMap column (particle size 2 μm, Thermo Scientific) was used for peptide separation. The LC gradient was set to be 3 h from 5% B to 30% B. The flow rate was 300 nl/min. Buffer A was composed of 2% acetonitrile, 0.1% formic acid in water, and buffer B was 2% water, 0.1% formic acid in acetonitrile.

In the MS method, MS2 spectra were acquired in the ion trap using CID mode. The normalized collision energy (CE) was set to be 35%. The Top 15-method was used, which means that the most abundant 15 precursors per cycle were selected for MS/MS analysis. We used the following resolution settings: MS1 recorded at 120,000 and MS2 recorded at normal resolution for trap-CID. Those MS1 signals of charge state +1 and those of unknown charge state were excluded from fragmentation and MS2 data generation. Precursor isolation width was 2 $m/z$. MS1 scans were set to a maximum of 1,000,000 counts and a maximum fill time of 200 ms. MS2 trap scans were set to a maximum of 10,000 counts and a maximum fill time of 100 ms. The dynamic exclusion parameters were set so that already sequenced precursors were excluded from reselection for 30 s.

Following measurement, the Elite data in.raw format was analyzed by Maxquant[55] and searched against human Swissprot database (downloaded April 2015) following standard SILAC settings. Specially, peptide tolerances at MS and MS/MS level were set to be 4.5 ppm and 0.75 Da, respectively. Up to two miss cleavages were allowed. Oxidation at methionine was set as a variable modification, whereas carbamidomethylation at cysteine was set as a fixed modification. To avoid incorrect identifications (especially for heavy channel) at early time points, the requantification and "match between runs" options were disabled. Other parameters are kept as default in Maxquant[55]. Both peptide- and protein-level results were controlled at 1% FDR[56].

**SWATH mass spectrometry.** Steady proteome and pSILAC samples were both measured by SWATH-MS. The same LC-MS/MS systems used for DDA measurements on SCIEX 5600 TripleTOF was also used for SWATH analysis[18,19]. For pSILAC samples, a 90-min LC gradient was used. For the MS method, we used a selection scheme of 64 variable wide precursor ion windows in the quadrupole. These windows were set to be at the ranges of 399.5–408.2, 407.2–415.8, 414.8–422.7, 421.7–429.7, 428.7–437.3, 436.3–444.8, 443.8–451.7, 450.7–458.7, 457.7–466.7, 465.7–473.4, 472.4–478.3, 477.3–485.4, 484.4–491.2, 490.2–497.7, 496.7–504.3, 503.3–511.2, 510.2–518.2, 517.2–525.3, 524.3–533.3, 532.3–540.3, 539.3–546.8, 545.8–554.5, 553.5–561.8, 560.8–568.3, 567.3–575.7, 574.7–582.3, 581.3–588.8, 587.8–595.8, 594.8–601.8, 600.8–608.9, 607.9–616.9, 615.9–624.8, 623.8–632.2, 631.2–640.8, 639.8–647.9, 646.9–654.8, 653.8–661.5, 660.5–670.3, 669.3–678.8, 677.8–687.8, 686.8–696.9, 695.9–706.9, 705.9–715.9, 714.9–726.2, 725.2–737.4, 736.4–746.6, 745.6–757.5, 756.5–767.9, 766.9–779.5, 778.5–792.9, 791.9–807, 806–820, 819–834.2, 833.2–849.4, 848.4–866, 865–884.4, 883.4–899.9, 898.9–919, 918–942.1, 941.1–971.6, 970.6–1006, 1005–1053, 1052–1110.6, 1109.6–1200.5. The MS2 acquisition mass range was set to 50–2000 $m/z$. For each window, optimized CE was applied. This was based on the calculation for a charge 2+ ion centered in the particular window, but with a spread of 15 eV. The dwell time was set to be 50 ms for all fragment ion scans (MS/MS scans) in "high-sensitivity" mode. In this way, a duty cycle was ~3.45 s consisting of a survey scan (high-resolution mode) of 250 ms and 64 fragment ion scans of 50 ms, respectively.

**SWATH-MS data analysis on protein expression.** For analyzing the steady-state proteomes of twin samples and unrelated samples, the SWATH-MS identification was performed by OpenSWATH software[24] searching against a previously established SWATH assay library, which contains mass spectrometric assays for 10,000 human proteins[26]. OpenSWATH first identified the peak groups from all individual SWATH maps at a target FDR = 1% (at least one run with $q$-value below 0.0236% for any given peak group) and then aligned between SWATH maps with extension FDR = 5% using a TRIC (transfer of identification confidence) algorithm that was specifically developed for targeted proteomic data analysis[25]. The

requantification feature in OpenSWATH was enabled, but only those peptide groups identified in more than 33% of the samples were proceeded for requantification. As discussed previously, protein FDR needs specific attention when the large, external spectral libraries are used in targeted proteomics[26]. Therefore, to pursue a strict quality control of protein-level identification, we further utilized Mayu software[27], which computed and applied an m_score cutoff of 0.00000218972 at the peptide peak group level to achieve the protein FDR of 1% in our final result. Using an in-house script, the signal normalization step by local intensity sums in retention time space[57] was performed at peptide level before summarizing protein intensities. To quantify the protein abundance levels across samples, we summed up the most abundant peptides for each protein (i.e., top three peptide groups based on intensity were used for those proteins identified with more than three proteotypic peptide signals, whereas all the peptides were summarized for other proteins). This allows for reliably estimating global protein level changes as shown in previous studies[58,59]. The protein expression data matrix was quantile normalized and $\log_2$ transformed for statistical and bioinformatics analysis.

**SWATH-MS data analysis on pSILAC data.** To analyze SWATH results of pSILAC experiment, we first generated a sample-specific library containing the light and heavy peptide assays. All the shotgun runs of the "Light" samples (those from steady proteome as well as $t_0$ samples in pSILAC experiment) were first searched against human SwissProt database using the iPortal pipeline[60]. Profile-mode.wiff files from shotgun data acquisition were centroided and converted to mzML format using the AB Sciex Data Converter v.1.3 and converted to mzXML format using MSConvert v.3.04.238. iPortal utilized iProphet schema[61] to integrate the search results from X!Tandem, Omessa, Myrimatch, and Comet at peptide level FDR = 1%. Xinteract option was "-dDECOY_ -OAPdlIw". Especially, peptide tolerances at MS and MS/MS level were set to be 75 ppm and 0.1 Da, respectively. Up to two missing trypsin cleavages were allowed. Oxidation at methionine was set as variable modification, whereas carbamidomethylation at cysteine was set as fixed modification. The light version of the raw spectral library was generated from all valid peptide spectrum matches and then refined into the non-redundant consensus libraries using SpectraST[62]. Using the spectrast2tsv.py function in Open-SWATH[24], we then generated both the light and heavy MS assays as the final library constructed from top 3–6 most intense fragments with Q3 range from 400 to 1200 $m/z$ excluding those falling in the precursor SWATH window were used for targeted data analysis of SWATH maps. The final library contained 41,386 peptide sequences of 4111 unique SwissProt proteins. OpenSWATH analysis was run with the same options as above for protein expression data, however, as the shotgun proteomic analysis, requantified data points were discarded for protein turnover calculation.

**Dimethylation shotgun proteomics.** The dimethylation experiment was conducted with the purpose to confirm the SWATH-MS results based on a pooling strategy, SAX fractionation[63], the following up shotgun measurement, and data analysis using Maxquant[55]. All reagents were purchased from Sigma-Aldrich (USA).

Dimethylation labeling based on "light" and "intermediate" channels was conducted following the in-solution labeling protocol[64]. Peptide digests from the three technical replicates (8 μg each) derived from both biological replicates T1DS and T2N twin samples were mixed and labeled with $CH_2O$ or $CD_2O$. The biological replicates were pooled again, respectively, for following SAX fractionation. The peptide digests from the two biological replicates of 11 unrelated healthy individuals and 11 DS cases (2 μg each) were also pooled, respectively, before dimethylation labeling reaction. Peptides labeled were first lyophilized, dissolved in ddH2O, and adjusted to pH 11 by adding 1 M NaOH before SAX fractionation.

Then, 50 μg of peptide mixtures (of twin samples and unrelated samples) were then separated into six fractions by SAX[63]. Totally six pH steps (11, 8, 6, 5, 4, 3, using buffer composed of 20 mM acetic acid, 20 mM phosphoric acid, and 20 mM boric acid titrated with NaOH to the desired pH) were applied and results in six SAX fractionations for further shotgun proteomic analysis.

The dimethylation peptide samples were measured on an EASY-nLC 1000 (Thermo Fisher) coupled to a Q Exactive Plus mass spectrometer (Thermo Fisher). Peptides were separated on a column (40 cm × 0.75 μm), packed in-house with reversed-phase ReproSil-Pur C18-AQ resin (1.9 μm, Dr. Maisch). Peptides were eluted for 80 min using a segmented linear gradient of 5–40% solvent B (99.9% acetonitrile, 0.1% formic acid) at a flow rate of 300 nl/min. Survey full-scan mass spectra were acquired with mass range 350–1500 $m/z$, at a resolution of 70,000 at 200 $m/z$ and the 20 most intense ions above an intensity of 7.3e4 were sequentially isolated, fragmented (HCD at 35 eV) and measured at a resolution of 17,500 at 200 $m/z$. Peptides with a charge of +1 or with unassigned charge state were excluded from fragmentation for MS2, and a dynamic exclusion of 30 s was applied. Ions were accumulated to a target value of 3e6 for MS1 and of 2e5 for MS2.

The raw files were searched against human SwissProt database using Maxquant. The "DimthLys0" and "DimethNter0" labels were enabled for the light labels and the "DimethLys4" and "DimethNter4" were enabled for heavy labels. Oxidation at methionine was set as variable modification, whereas carbamidomethylation at cysteine was set as the fixed modification. Up to two missed cleavages were allowed.

The peptide tolerances were set as 6 and 20 ppm at MS and MS/MS levels. The feature of "match between runs" and requantification were enabled with a match window of 0.7 min. Other parameters are kept as Default in Maxquant[55]. Both peptide- and protein-level results were accepted at 1% FDR.

Specifically, to test the labeling efficiency, the "DimthLys0" and "DimethNter0" labels and the "DimethLys4" and "DimethNter4" labels were, respectively, set as variable modifications for the "light" and "Intermediate" samples. Accordingly, we determined the labeling efficiency is 99.66 and 99.86% for the "light" and "heavy" channels based on the numbers of total peptides modified by dimethylation labels or not. Further among the peptides ending with lysine, 99.84 and 99.90% of them were dimethyl labeled at both peptide N terminus and the side chain of lysine residues. The data suggested the dimethyl labeling was completed in this experiment.

**Determination of protein turnover rate.** In pSILAC, the turnover behavior of heavy and light signals of different proteins thus permits the quantification of protein-specific degradation rates ($K_{deg}$). The rates of loss of the light isotope ($k_{loss}$) were directly calculated from the output data matrix generated by OpenSWATH. We modeled the relative isotope abundance (RIA)[21], defined as the signal intensity in the light channel divided by the sum of light and heavy intensities, onto an exponential decay model assuming a null heavy intensity (RIA = 1) at time 0, i.e.,

$$RIA(t) = e^{(-K_{loss} \times t)}. \tag{1}$$

The fit was obtained through nonlinear least-squares estimation ("nls" function in R, with a starting $K_{loss}$ parameter of 0.1).

To calculate protein level $K_{loss}$, when several peptides were available for a protein we first excluded peptides whose signal was absent in some of the samples. When more than four peptides were available for a certain protein, we excluded outliers (defined as having a probability <10e−4 in the empirical distribution of median $K_{loss}$ of the peptides of that protein). We then performed a weighted average of the $k_{loss}$ value of each peptides of the protein, using as weights the number of data points available for the peptide divided by the variance of peptide's $k_{loss}$ estimate (obtained from the model's fit), which ensured giving more weight to peptides carrying robust information. Importantly, this method maximized correlation and minimized median absolute error across replicates.

Next the final protein degradation rate $K_{deg}$ was calculated by the equations[21,22]:

$$K_{deg} = K_{loss} - K_{dil} \tag{2}$$

$$K_{dil} = \ln(2)/t_{cc}, \tag{3}$$

where $K_{dil}$ is the dilution rate due to the cell doubling and $t_{cc}$ denotes the cell doubling time (which differs between T1DS and T2N).

**Absolute label-free quantification.** The practice of aLFQ was used[20]. In the present study, to derive absolute protein abundance estimates, 34 anchor proteins covering a wide abundance range in human proteome were targeted by manual data extraction using Skyline[65]. A total of 42 heavy reference peptides with known amount were spiked into a representative T1DS and a T2N sample to determine the endogenous levels of peptides digested from the samples, by absolute quantification (AQUA)[66]. The manual inspection of the light and heavy peaks using Skyline excluded low intensity and low-quality peak groups and also exclude the light-to-heavy ratios below 0.04 or above 25, resulting in reliable quantification of 20 final anchor proteins by 20 AQUA peptides. In this way, the absolute moles of these anchor proteins were obtained.

In the next step, we averaged the total protein concentrations quantified by Bio-Rad assay from the same amount of cells (1 million) of all the fibroblast samples (regardless of T1DS or T2N, or unrelated cases). This averaged constant links the number of cells being digested to the micrograms of total proteins (i.e., it provides a general total protein concentration per cell). Thus, for those final anchor proteins whose absolute moles were already quantified (per 1 μg protein digest injected into LC-MS), we estimated the copies per cell numbers for all the anchor proteins. Next, based on the log-log correlation (Supplementary Fig. 10) between the SWATH-MS intensities (from protein expression data of one T1DS and T2N sample) and the copies per cell values for all the anchor proteins, we generated the linear equation for T1DS and T2N status separately. Finally, the copies per cell values (absolute amounts) of all the proteins in the proteome quantified in "protein expression analysis" can be determined using the respective log-log linear equation and their measured SWATH-MS intensities in T1DS and T2N.

In the above approach, we did not apply the aLFQ model selection[67] and we performed the AQUA only based on single-point ratios (see above) rather than dilution curves. Although the estimation accuracy of absolute protein concentration per cell might be lower than the previous studies which optimized these steps[20,67], our simple approach here essentially provided the copies per cell estimate for all the proteins (including those quantified with only one peptide), and enabled the proteome-wide investment comparison between T1DS and T2N. Another consideration is the possible deviation of the protein amount per cell estimation due to the variations during protein extraction, concentration determination, and peptide digestion. However, such a deviation is unlikely to

impair the present study as parallel protein extraction and digestion was performed between samples.

**Estimating protein synthesis expense and proteome investment**. We determine the protein synthesis rate based on following equations and reasoning[29,68]. For a state in equilibrium, the relationship between the rate constants for synthetic and degradative reactions are can be described as

$$\mathrm{d}Px/\mathrm{d}t = K_{\mathrm{syn}} + P \times e^{\left(-K_{\mathrm{deg}} \times t\right)}, \qquad (4)$$

where $\mathrm{d}Px/\mathrm{d}t$ denotes the change of protein amount in time, and $Px(t)$ is the absolute amount of the protein x at time $t$, which is estimated by the aLFQ in the present study.

At a steady state, $\mathrm{d}Px/\mathrm{d}t = 0$. Therefore,

$$K_{\mathrm{syn}} = Px(t)K_{\mathrm{deg}} \qquad (5)$$

i.e., in a given cell, $K_{\mathrm{syn}} =$ protein copies (i.e., quantile normalized copies for generating Fig. 5f) times $K_{\mathrm{deg}}$.

We further estimated the protein investment for each protein[69]. The total mass of a particular protein was calculated by multiplying number of instances of the protein (i.e., protein absolute quantities) by the mass of the individual proteins (i.e., protein MW) was used as a reference for the cost associated to the production of each protein. Accordingly the protein synthesis expense was calculated as $K_{\mathrm{syn}}$ multiplied by the mass of individual proteins.

The above parameters were combined with the GO biological annotations to infer the ratio of proteome investment and protein synthesis expenses between organelles.

**Functional annotation and enrichment analysis**. To annotate protein complexes, we assembled a list of known stable protein complexes and then annotated proteins as either being part of these protein complexes ("complex in", those proteins involved in protein complex according to either CORUM or reactome databases[32,33]) or not ("complex out", other proteins). Note that the "complex out" group may contain proteins that are actually part of (yet unknown) protein complexes and that the "complex in" group may contain proteins that not actually assembled in fibroblasts under the conditions we tested. Despite these limitations, we assumed that these two groups were enriched for bona fide protein complex and non-protein complex proteins, respectively.

The GSEA was performed by the online toolkit based on http://babelomics.bioinfo.cipf.es/ in which the list of proteins (in SwissProt IDs) were ranked with the T1DS/T2N FC values of transcript, protein, and degradation, and then a logistic model is used to detect gene sets (functional annotations of GO biological processes) that are consistently associated to high or low values in the ranked lists[35]. The adjusted $P$-value $< 0.05$ from the logistic regression was used to determine the statistical significance of GSEA analysis. A total of 651, 1020, and 250 biological processes were identified to be significant at mRNA, protein, and protein degradation levels. The logarithm odds ratio (LOR) comparing two groups or experimental conditions was reported for each processes, LOR $> 0$ indicates a functional term is over-represented for genes upregulated in T1DS and LOR $< 0$ indicates a functional term is over-represented for genes downregulated in T1DS.

DAVID Bioinformatics Resources 6.7 (https://david.ncifcrf.gov/)[70] was used to perform the enrichment analysis of GO biological processes and cellular component for all the five segments evenly divided from the T1DS/T2N FC of protein degradation and to report $P$-values of enrichment (by a modified Fisher exact test). Briefly, all the identified 4056 proteins were taken as the background for comparison of proteins in each segment. The $P$-values of manually compiled processes showing significance ($P < 0.05$) in any of the segments were used to indicate the relative enrichment pattern between segments, with a minimum of four proteins per category. If one category were annotated with $<4$ proteins in a segment, statistical insignificance was assigned ($P = 1$) for the visualization.

For the further mapping analysis of GO subcellular annotation, bioMart (http://www.biomart.org/) was used to generate the cellular component for all human proteins and then filtered based on the identified protein list. All the offspring GO items to each organelle were included in the annotation. Those proteins annotated at multiple cellular components were all accepted and included for the analysis of each respective component.

**Statistics and visualization**. We used the boxplot function (R package "graphics", default parameters) to generate all the box plots. The box denotes the range of 25–75th percentages. The bold line in the box represents the median of the data sets. The whiskers are defined as $\min(\max(x), Q_3 + 1.5 \times IQR)$ for the upper whisker and lower whisker: $\max(\min(x), Q_1 - 1.5 \times IQR)$ for lower whisker. Here, $Q_1$ and $Q_3$ represent the 25th and 75th quantiles, respectively, and $IQR = Q_3 - Q_1$. Outliers were defined based on whisker positions. Wilcox test (wilcox.test in R) was used to infer $P$-values.

**Networks linking Chr21 proteins and regulated processes**. Protein expression values were log2 transformed. For the unrelated samples, the average of the two measurements in the replicates was calculated for each protein. Next, a FC was

calculated for each protein by (i) obtaining the median value for the measurements in normal and in trisomy samples and then (ii) subtracting the median of normal values from the median of trisomy values. This was done separately for twins and unrelated samples. In addition, proteins whose quantities strongly varied between measurements were excluded from further calculations. For this, original (non-transformed) values of the healthy samples were median-centered and proteins for which the ratio of the median absolute deviation over median exceeded 0.5 across measurements were considered to have inherent quantitative variability. This calculation was done independently for twins and unrelated samples. Proteins variable in the twins data set were excluded from the calculations there, and proteins variable in either set were excluded from the top hits in the unrelated samples. Next, a set of the most highly deregulated proteins in the disease samples was first defined as top 5% proteins with the highest and top 5% proteins with the lowest FC in the twins samples (406 proteins in total). A set of the same size with top and bottom 203 proteins with the strongest FCs was then also composed for the unrelated samples. These two sets were further extended with all measured proteins from the Chr21. Proteins in the two sets were then annotated with the reactome pathways they were associated with. Enrichment of particular pathways in each set was calculated by assessing the over-representation of the term compared to the background of all other measured proteins (excluding highly variable ones when appropriate). Fisher exact test was used to obtain the associated $P$-values and these were further corrected for multiple testing by applying the Benjamini–Hochberg correction.

The Fig. 4 depicts the reactome process enriched among the highly up- and downregulated proteins in either the twins or unrelated sets. Only the processes with the corrected $P$-value $<0.01$, containing three or more proteins and having enrichment higher than 2 over the background proteins are shown. For the unrelated samples, only two processes passed these criteria; all other proteins depicted on the figure represent those strongly deregulated proteins in the unrelated samples, which are annotated with the terms enriched in the twins samples selected according to the above criteria. When the same group of proteins was associated with two or more significant reactome terms, the term having a smaller $P$-value or containing significantly more proteins from the list was selected as the representative one. Several of the proteins were still annotated with two or more of the selected terms. In these cases, they are colored according to the term with the lowest $P$-value. The names of the proteins encoded on the Chr21 are shown in red—they were added to the sets even if they were not in the top 5% deregulated proteins. The border on the circle denotes the direction of deregulation: upregulated proteins are shown with the black border and downregulated ones with the light gray border.

**Western blots**. One-dimensional polyacrylamide gel electrophoresis (1D-PAGE) was performed by using precast NuPAGE Novex 4–12% Bis–Tris gels (Invitrogen, Switzerland). About 15 μg of protein was loaded onto each well for the cell lysate. Proteins were blotted onto a polyvinylidene fluoride (PVDF) membrane (iBlot Dry Blotting System, Invitrogen), and membranes were blocked with 5% (w/v) non-fat dry milk in TBST buffer (tris-buffered saline, 0.1% Tween 20) for 1 h at room temperature and incubated with total OXPHOS Rodent WB Antibody Cocktail (#ab110413, Abcam, UK) in diluted concentration of 1:1000 at 4 °C overnight. This antibody cocktail contains five mouse monoclonal antibodies, one each against CI subunit NDUFB8 (#ab110242, Anti-NDUFB8 antibody), CII-30 KDa (#ab14714, Anti-SDHB antibody), CIII-Core protein 2 (#ab14745, Anti-UQCRC2 antibody), CIV subunit I (#ab14705, Anti-MTCO1 antibody), and CV alpha subunit (#ab14748, Anti-ATP5A antibody), all from Abcam, UK. Anti-Actin antibody (#ab3280) was used as a loading control and also purchased from Abcam. Membranes were washed four times for 15 min with TBST buffer and subsequently incubated with a secondary antibody, Amersham ECL Mouse IgG, HRP-linked whole Ab (#NA931-100UL, GE Healthcare) at dilutions 1:20,000. The final Western blot signals were developed with an Amersham ECL Prime Western Blotting Detection Reagent (GE Healthcare), by chemiluminescence using darkroom development techniques. Please see uncropped scans of Western blots in Supplementary Fig. 19.

**Data availability**. All the raw data of mass spectrometry measurements, together with the input spectral library and OpenSWATH results can be freely downloaded from ProteomeXchange Consortium (http://proteomecentral.proteomexchange.org) via identifier PXD004880. Other data are available from the corresponding authors upon reasonable request.

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

## Acknowledgements

We acknowledge contributions by Ben Collins, Rodolfo Ciuffa, Hannes Rost, George Rosenberger, Sandra Goetze, Christian Feller, Federico Santoni, Geogios Stamoulis to the study. Y.L. was supported by the European Research Council (ERC, consolidator grant number 616441-DISEASEAVATARS). The group of SEA was supported by grants for the ERC AdG249968, SNF 163180, and the ChildCare Foundation. The group of RA was supported by ERC grants Proteomics v3.0 (AdG-233226 Proteomics v.3.0) and AdG-670821 Proteomics 4D), and the Swiss National Science Foundation (SNSF) grant number: 31003A_166435. M.B. is funded by SystemsX.ch. A.B. and L.L. were supported by the German Federal Ministry of Education and Research (BMBF; grants: Sybacol & PhosphoNetPPM). The group of G.T. was supported by the European Research Council (ERC grant number 616441-DISEASEAVATARS), the Umberto Veronesi Foundation (fellowship to P.-L.G.), the ERA-NET Neuron Program (G.T. and P.-L.G.), Regione Lombardia (Ricerca Indipendente 2012), and Italian Ministry of Health (Ricerca Corrente to G.T.)

## Author contributions

Y.L., C.B., S.E.A. and R.A. designed the experiments. Y.L. conducted most of the proteomic experiments with assistance from T.M. and P.J.B. C.B. and S.E.A. established the genetic fibroblast cells. Y.L. and L.L. performed the main bioinformatics analyses. E.G.W., P.-L.G. and M.B. advised on mass spectrometry data interpretation and visualization. T.S. performed the Western blot experiment. A.B., E.G.W., M.F., W.S. and G.T. provided critical input to the writing. Y.L., A.B., S.E.A., R.A. wrote the paper with input from all authors. A.B., S.E.A., R.A. supervised the study.

## Additional information

**Competing interests:** The authors declare no competing financial interests.

