## [Peer Review File · Nature Communications]

REVIEWERS' COMMENTS:

Reviewer #1 (Remarks to the Author):

Overall the authors have done a nice job in responding to my comments. I believe that this manuscript should be considered as being "competitive" for being published in Nature Communications.

I have only 2 remaining comments, both were raised in the original round of reviews.

1. It would have been nice to see additional validation / follow-up of a few candidate proteins identified in the proteomic studies. The authors do provide a bit more data and a data comparison to previous publications which helps. I am unsure how Nature Communications handles this sort of situation.
2. The use of fibroblasts limits the impact of the results but does not invalidate them. Changing the title does reduce this limitation but does not solve it.

Reviewer #2 (Remarks to the Author):

This manuscript was reviewed previously at [Redacted]. The authors elected to move the manuscript to Nature Communications where it is a better fit. I found the responses to my comments to be sufficient and I am now ready to recommend this paper for publication.

Reviewer #1 (Remarks to the Author):

Overall the authors have done a nice job in responding to my comments. I believe that this manuscript should be considered as being "competitive" for being published in Nature Communications.

I have only 2 remaining comments, both were raised in the original round of reviews.
1. It would have been nice to see additional validation / follow-up of a few candidate proteins identified in the proteomic studies. The authors do provide a bit more data and a data comparisons to previous publications which helps. I am unsure how Nature Communications handles this sort of situation.

> We thank for the generally positive comments. We agree that additional validations are always nice for Proteomics/ Systems Biology publications. In fact we believe we did a great job here compared to many other proteomic papers published (as discussed in the last round revision): We used different levels of verifications in this paper a) the validation using unrelated samples by including the individual genetic variation for all the conclusions, and b) the validation using di- methylation labeling based shotgun proteomics experiments for all the relevant SWATH-MS results. Please kindly note that this is normally not done in other proteomics studies, for a well-established mass spectrometric method like SWATH-MS. c) Western blots based validation for mitochondria dysregulation. d) Inclusion of and comparison to other published data sets, as illustrated in Figure 2c and Supplementary Figure 5.

2. The use of fibroblasts limits the impact of the results but does not invalidate them. Changing the title does reduce this limitation but does not solve it.

> This point regarding fibroblast cells is well summarized here by the reviewer. We have extensively addressed this question in the last revision. The biggest effort we made was to focus on fibroblast cell type, which is a very important cell line in DS research (see the Supplementary Table 1 in this version-which was previously Supplementary Table 7, and please also see our reply in the last round of revision). And we do not have further things to add here, but to add this note to Page 14, Para 1 in the discussion "... Nevertheless, the use of fibroblast may limit the impact of certain findings present and the study of other cell types and tissues will be necessary to fully understand the specific effects and other phenotypes of T21..."

Reviewer #2 (Remarks to the Author):

This manuscript was reviewed previously at [Redacted]. The authors elected to move the manuscript to Nature Communications where it is a better fit. I found the responses to my comments to be sufficient and I am now ready to recommend this paper for publication.

> Thanks.